# Machine Learning Analysis in Diffusion Kurtosis Imaging for Discriminating Pediatric Posterior Fossa Tumors: A Repeatability and Accuracy Pilot Study

**DOI:** 10.3390/cancers16142578

**Published:** 2024-07-18

**Authors:** Ioan Paul Voicu, Francesco Dotta, Antonio Napolitano, Massimo Caulo, Eleonora Piccirilli, Claudia D’Orazio, Andrea Carai, Evelina Miele, Maria Vinci, Sabrina Rossi, Antonella Cacchione, Sabina Vennarini, Giada Del Baldo, Angela Mastronuzzi, Paolo Tomà, Giovanna Stefania Colafati

**Affiliations:** 1Oncological Neuroradiology and Advanced Diagnostics Unit, Bambino Gesù Children’s Hospital, IRCCS, 00165 Rome, Italy; paul.voicu@hotmail.it (I.P.V.); francesco.dotta@opbg.net (F.D.); eleonora.piccirilli@opbg.net (E.P.); claudia.dorazio@opbg.net (C.D.); 2Department of Innovative Technologies in Medicine and Dentistry, University G. d’Annunzio of Chieti-Pescara, 66100 Chieti, Italy; 3Medical Physics Unit, Bambino Gesù Children’s Hospital, IRCCS, 00165 Rome, Italy; antonio.napolitano@opbg.net; 4Department of Neuroscience, Imaging and Clinical Sciences, University G. d’Annunzio of Chieti-Pescara, 66100 Chieti, Italy; massimo.caulo@unich.it; 5Neurosurgery Unit, Bambino Gesù Children’s Hospital, IRCCS, 00165 Rome, Italy; andrea.carai@opbg.net; 6Onco-Hematology, Cell Therapy, Gene Therapies and Hemopoietic Transplant, Bambino Gesù Children’s Hospital, IRCCS, 00165 Rome, Italy; evelina.miele@opbg.net (E.M.); antonella.cacchione@opbg.net (A.C.); giada.delbaldo@opbg.net (G.D.B.); angela.mastronuzzi@opbg.net (A.M.); 7Paediatric Cancer Genetics and Epigenetics Research Unit, Bambino Gesù Children’s Hospital, IRCCS, 00165 Rome, Italy; maria.vinci@opbg.net; 8Pathology Unit, Bambino Gesù Children’s Hospital, IRCCS, 00165 Rome, Italy; sabrina2.rossi@opbg.net; 9Pediatric Radiotherapy Unit, IRCCS Fondazione Istituto Nazionale Tumori, 20133 Milano, Italy; sabina.vennarini@istitutotumori.mi.it; 10Radiology and Bioimaging Unit, Bambino Gesù Children’s Hospital, IRCCS, 00165 Rome, Italy; paolo.toma@opbg.net

**Keywords:** child, magnetic resonance imaging, diffusion tensor imaging, posterior fossa tumors, machine learning

## Abstract

**Simple Summary:**

Differentiating pediatric posterior fossa (PF) tumors such as medulloblastoma (MB), ependymoma (EP), and pilocytic astrocytoma (PA) remains relevant, due to important treatment and prognostic implications. Diffusion kurtosis imaging (DKI) has not been tested to date to distinguish between pediatric PF tumors. Estimating diffusion values from whole-tumor-based (VOI) segmentations may improve the repeatability of diffusion measurements compared to conventional region-of-interest (ROI) approaches. Our purpose was twofold: to assess the repeatability of VOI DKI-derived diffusion measurements and DKI accuracy in discriminating among pediatric PF tumors, by employing conventional statistical analyses and machine learning (ML) techniques.

**Abstract:**

**Background and purpose**: Differentiating pediatric posterior fossa (PF) tumors such as medulloblastoma (MB), ependymoma (EP), and pilocytic astrocytoma (PA) remains relevant, because of important treatment and prognostic implications. Diffusion kurtosis imaging (DKI) has not yet been investigated for discrimination of pediatric PF tumors. Estimating diffusion values from whole-tumor-based (VOI) segmentations may improve diffusion measurement repeatability compared to conventional region-of-interest (ROI) approaches. Our purpose was to compare repeatability between ROI and VOI DKI-derived diffusion measurements and assess DKI accuracy in discriminating among pediatric PF tumors. **Materials and methods**: We retrospectively analyzed 34 children (M, F, mean age 7.48 years) with PF tumors who underwent preoperative examination on a 3 Tesla magnet, including DKI. For each patient, two neuroradiologists independently segmented the whole solid tumor, the ROI of the area of maximum tumor diameter, and a small 5 mm ROI. The automated analysis pipeline included inter-observer variability, statistical, and machine learning (ML) analyses. We evaluated inter-observer variability with coefficient of variation (COV) and Bland–Altman plots. We estimated DKI metrics accuracy in discriminating among tumor histology with MANOVA analysis. In order to account for class imbalances, we applied SMOTE to balance the dataset. Finally, we performed a Random Forest (RF) machine learning classification analysis based on all DKI metrics from the SMOTE dataset by partitioning 70/30 the training and testing cohort. **Results**: Tumor histology included medulloblastoma (15), pilocytic astrocytoma (14), and ependymoma (5). VOI-based measurements presented lower variability than ROI-based measurements across all DKI metrics and were used for the analysis. DKI-derived metrics could accurately discriminate between tumor subtypes (Pillai’s trace: *p* < 0.001). SMOTE generated 11 synthetic observations (10 EP and 1 PA), resulting in a balanced dataset with 45 instances (34 original and 11 synthetic). ML analysis yielded an accuracy of 0.928, which correctly predicted all but one lesion in the testing set. **Conclusions**: VOI-based measurements presented improved repeatability compared to ROI-based measurements across all diffusion metrics. An ML classification algorithm resulted accurate in discriminating PF tumors on a SMOTE-generated dataset. ML techniques based on DKI-derived metrics are useful for the discrimination of pediatric PF tumors.

## 1. Introduction

Brain tumors are the most frequent solid tumors in children [1]. A significant fraction of pediatric brain tumors (estimated between 45% and 60% [2]) are located in the posterior cranial fossa (PF). Among PF tumors, medulloblastoma (MB), pilocytic astrocytoma (PA), and ependymoma (EP) are the most common lesions, with reported prevalences of 30% to 40%, 25% to 35%, and 10% to 15%, respectively [3]. The correct pre-operative differentiation between these lesion subtypes may have important therapeutic implications, since surgical treatment of PF tumors may be often associated with significant morbidity including cerebellar mutism [4], which has been significantly correlated to tumor size, location, and histology.

MRI is the imaging modality of choice for the noninvasive diagnosis of pediatric PF tumors. In this regard, diffusion-weighted imaging was investigated and deemed promising for both discriminating among PF tumor subtypes [3,5,6,7] and tumor grading. To obtain the information from MRI sequences (including diffusion sequences), the tissue of interest—in this case the tumor—must be segmented (or annotated). Segmentation can be performed on a region-of-interest (ROI) or volume-of-interest (VOI) basis and can be manual, semi-automated, or automated. ROI segmentation is usually performed on a single slice and can be small or large, as needed. Historically, ROI segmentations performed manually were the first means to collect information from MRI images. VOI segmentation entails annotating the whole tumoral volume, i.e., selecting the whole tumor on all slices where it is observed. If performed manually, VOI segmentation requires significantly more time than ROI segmentation and, probably for this reason, was less employed in older studies. In previous studies on pediatric PF tumors, both region-of-interest (ROI) and whole-tumor volume-based (VOI) segmentation approaches have been used. However, it is unclear which of the two should be used for pediatric PF tumors. ROI-based measurements have been historically preferred, due to their ease of implementation in a clinical setting and faster acquisition if the segmentation task is performed manually. In theory, however, ROI-based measurements could be hindered by higher inter-observer variability, thus reducing measure repeatability, i.e., extracting the same results from two different segmentations performed by two different readers. Even if most studies showed good to excellent ROI-based accuracy [3,5,6,7], it is still unclear whether extracting diffusion metrics from more tumor tissue with a whole-tumor volume-based (VOI) segmentation instead of an ROI-based segmentation could benefit measure repeatability for PF tumors. This aspect may prove relevant because of the current availability of free segmentation tools [8,9] that allow obtaining tumor VOI in a limited, clinically feasible, amount of time.

In recent years, studies based on machine learning (ML) classifiers have been tested for discriminating pediatric PF tumors [10], among tumor subtypes such as MB and EP [11], and for texture characterization to develop a radiomic signature of PF ependymomas [12]. Most of these studies were based on conventional MRI sequences, although some also incorporated ADC maps into their analysis. One study from Novak et al. specifically tested diffusion-weighted imaging-based ML in a multicenter retrospective evaluation [13].

Diffusion-weighted imaging (DWI), first introduced in 1985 by Denis LeBihan, measures water diffusion in normal and pathological conditions in different organs and tissues, including the central nervous system. From a technical standpoint, DWI is based on a slight modification of T2-weighted sequences, where two diffusion gradients of equal strength (termed the b values) are applied in opposing directions. This allows for estimating the magnitude of water diffusion (expressed in mm^2^/s) by obtaining apparent diffusion coefficient (ADC) maps. DWI was the first MRI imaging technique using physiology to obtain insights into pathological brain conditions such as stroke, abscess, and tumors. DWI has had wide success in its clinical application and is routinely employed in all imaging protocols studying the brain [14].

An extension of DWI, diffusion tensor imaging (DTI), was first been described by Basser in 1994 [15] and subsequently developed with Pierpaoli [16]. DTI derives its name from the mathematical object tensor [15,16] and was developed to better estimate water diffusion in anisotropic, heterogeneously oriented tissues compared to ADC, which was limited by measuring diffusion in only two relevant directions. Using the six scalar elements of the diffusion tensor, besides mean diffusivity (MD, the equivalent of the apparent diffusion coefficient), it is also possible to calculate the rotationally invariant metric fractional anisotropy (FA), FA has been useful to characterize noninvasively the white matter tracts in the brain, which are useful for lesion characterization and neurosurgical planning.

Diffusion kurtosis imaging (DKI) is a versatile diffusion technique [17] that allows obtaining metrics based on a non-Gaussian model of water diffusion (such as mean kurtosis, radial kurtosis, and axial kurtosis) but also allows obtaining typical DTI metrics based on a Gaussian model of water diffusion such as fractional anisotropy or mean diffusivity (the equivalent of the apparent diffusion coefficient). From a mathematical standpoint, DKI is based on the dimensionless metric kurtosis (see also Materials and Methods, Section 2.2). DKI has shown promise for adult glioma grading [18] and has been analyzed for normal pediatric brain development [19]. One of the limits of DKI is its long acquisition time (in the order of approximately 15 min). This issue was partially mitigated by the development of multi-slice (or multiband) techniques [20] for DKI acquisition, which allowed reducing the required scanning time for the sequence. Recently, our group first studied DKI in pediatric gliomas [21], proving DKI accurate in their grading. In addition, the DKI-based model proved to be a very accurate predictor of patient overall and progression-free survival. In the same paper, ML techniques based on the synthetic minority oversampling technique (SMOTE) were applied to exclude the results due to the imbalanced dataset [21].

In this study, we analyzed the repeatability of ROI and VOI DKI-derived diffusion measurements and explored whether DKI was accurate in discriminating among pediatric PF tumors with ML techniques.

## 2. Materials and Methods

### 2.1. Patients

This single-institution retrospective study was approved by our Internal Review Board and was compliant with the Declaration of Helsinki. We identified 34 consecutive patients with posterior fossa brain tumors and available histological diagnosis who underwent preoperative MRI including DKI imaging. Lesions were classified according to the 2021 WHO classification [22].

### 2.2. Diffusion Kurtosis Imaging (DKI)

In 2005, Jensen and Helpern first described the mathematical basis, which allows DKI implementation [23]. The diffusion of water through biological tissues can be considered as a random process, which is statistically described by a probability distribution. In the simplest models (such as DTI), this distribution is assumed to be Gaussian. However, in biological tissues that may possess a complex structure, the diffusion displacement probability distribution may deviate substantially from a Gaussian form. This deviation from Gaussian behavior can be quantified using a statistical metric called kurtosis. Kurtosis can be thought of as a measure describing the amplitude and shape of a distribution tail compared to its overall shape (Figure 1). When high kurtosis is present, distributions typically have more tail data than normally distributed data (platykurtic distribution) and vice versa.

DKI is a diffusion-weighted MRI technique that allows the diffusional kurtosis to be estimated with clinical scanners using standard diffusion-weighted pulse sequences. DKI is an extension of the more widely used DTI model, which requires the use of at least 3 b-values and 15 diffusion directions. The degree of non-Gaussianity by the diffusional kurtosis and derivative metrics, such as the mean, axial, and radial kurtosis, can therefore be measured.

### 2.3. Image Acquisition, Preprocessing, and Analysis

All patients underwent a standardized MRI protocol on a 3 Tesla magnet (Siemens, Erlangen, Germany), including DKI sequences. Imaging under sedation was obtained when the patients were unable to cooperate.

DKI data acquisition was based on a prototype EPI-based diffusion-weighted sequence with blipped controlled aliasing for simultaneous multi-slice. Diffusion measurements were performed with dipolar diffusion sensitizing gradients applied in 30 directions with b values of 0 (10 averages), 1000, and 2000 s/mm^2^.

For each patient, we implemented a standardized post-processing DKI model. The diffusion data analysis pipeline included motion correction [24,25] performed via artifact correction in diffusion MRI (ACID) Matlab toolbox (Matlab 2021b, http://www.diffusiontools.com/ (accessed on 21 September 2022)), whereas denoising, brain masking, and diffusion kurtosis tensor estimation were performed using Mrtrix (version 3.0.4, http://www.mrtrix.org (accessed on 3 October 2022)). DKI-derived metrics estimation was computed as described previously [25]. All processing and metric estimations were then embedded in batch Matlab function (Matlab 2021b, Mathworks, Natick, MA, USA). The standardized pipeline output included 5 metrics: MK, AK, RK, FA, and MD. Conventional sequences were then registered to DKI maps using an affine transformation tool in 3DSlicer.

### 2.4. Region-of-Interest (ROI) and Volume-of-Interest (VOI) Measurements

For each patient, two neuroradiologists (IPV and GSC with 9 and 25 years of experience, respectively) independently segmented the whole-solid tumor volume, the area of maximum tumor diameter, and a small polygonal ROI measuring 5 mm (21 voxels, 168 mm^3^ volume) in the region with highest DKI values. Segmentations were performed with the free tool ITK-SNAP [8,9]. Whole-tumor volumes (VOIs) were segmented semi-automatically by using tissue classification and the active contour features in ITK-SNAP. The maximum area ROI was processed automatically from each reader volumetric segmentation with a Matlab in-house script. Large ROIs approximated between 150 and 250 voxels (1200–2000 mm^3^ volume), while VOIs approximated between 1250 and 2500 voxels (10,000–20,000 mm^3^ volume). Small ROIs were placed manually. For small ROI segmentations, the neuroradiologists selected the region with the highest DKI values instead of the region with mean values, in order to reduce possible biases related to selecting a small area in lesions with significant intra-tumoral heterogeneity. Cystic tumor portions measuring more than 2 mm, calcifications, and hemorrhagic components were avoided. DKI-derived metrics values were normalized to the normal-appearing white matter of the cerebral hemisphere unaffected by the tumor. Normal-appearing white matter was segmented with the same method as the tumor for each type of segmentation. Neuroradiologists were blinded to the other neuroradiologist segmentation and histological diagnosis.

### 2.5. Statistical Analysis

In order to assess whether the quantitative results obtained from the DKI metrics were repeatable and whether the type of segmentation could influence measurer repeatability, the segmentation results from the first neuroradiologist were compared to the results of the segmentation of the other neuroradiologist, by measuring interobserver variability for each segmentation type and each metric. Interobserver variability was assessed with a coefficient of variation (COV) and Bland–Altman plots.

The coefficient of variation (COV) is a robust standardized measure of distribution and is expressed as the ratio between the standard deviation and the mean. COV is widely used in the fields of analytical chemistry, engineering, or physics to express the precision and repeatability of measurements, including diffusion-related metrics [26].

Bland–Altman plot is a well-known method of data plotting, used in biosciences to express graphically the agreement between two measurements [27].

DKI values’ accuracy in discriminating among tumor histology in posterior fossa tumors was assessed with multivariate analysis of variance (MANOVA, Pillai’s trace) and post hoc *t* tests with Benjamini–Hochberg correction for multiple comparisons.

### 2.6. Machine Learning Analysis with SMOTE and Random Forest Classifier

In order to exclude that statistical results of the analysis may have been influenced by an imbalance between different subgroups of the dataset, we performed further testing with the random synthetic minority oversampling technique (SMOTE) analysis (package SMOTE from the library imblearn (version 0.12.3) in Python, version 3.9.12). The SMOTE preprocessing algorithm is considered “de facto” the standard in the framework of learning from imbalanced data [28]: It carries out an oversampling approach to rebalance the original training set by introducing new synthetic examples. The procedure is focused on the “feature space” rather than on the “data space”, in other words, the algorithm is based on the values of the features and their relationship, instead of considering the data points as a whole [29].

We choose to construct the random forest classifier by testing various hyperparameters and selecting the ones with the best predictive performance on the training set.

Finally, we selected a random seed to guarantee the reproducibility of the results among the different computers.

Using SMOTE, we generated a total of 11 synthetic observations (10 of ependymoma and 1 of pilocytic astrocytoma), resulting in a dataset with 45 instances (34 original and 11 synthetic). Thereafter, machine learning classification analysis was performed in the Python environment (version 3.9) using all the DKI metrics in the dataset obtained by the SMOTE technique. The dataset was randomly split into 70% for the training test and 30% for the testing set.

The Random Forest classifier was implemented in the machine learning analysis, and it was initially trained in the training set and then evaluated in the testing set through common statistical measures such as accuracy and area under the curve ROC (AUC). We evaluated the classifier performance in 300 different bootstraps, to get a better idea of the classifier performance in slightly different datasets. In this way, we built a confidence interval regarding the performance of the classifier in the discrimination of each FCP tumor type (MB, PA, EP).

Statistical analyses were performed, with Excel 2011 (Microsoft, Redmond, WA, USA), SPSS (version 20), R Studio (version 1.1.463), and Python (version 3.9). *p* values < 0.05 were considered statistically significant. The pipeline for the analysis can be accessed at the GitHub repository https://github.com/Francescodotta/Posterior_Fossa_Tumors_DKI/blob/main/Posterior_Fossa_DKI_Pipeline_Analysis.ipynb (accessed on 22 December 2022).

## 3. Results

Histological diagnosis was available in all 34 patients, including medulloblastoma (15 patients), pilocytic astrocytoma (14 patients), or ependymoma (5 patients). The mean age at diagnosis was 7.48 years (95% CI: 4.52–10.45).

COV between the two observers for segmentations of tumor volume, maximum area, and small ROI, respectively, are reported in Table 1 and Figure 2. COV was constantly lower for volume segmentation than the other two types of segmentation for each metric. Maximum area ROI measurements also presented lower COV than small ROI measurements for each metric.

Bland–Altman plots of inter-observer variability between the two neuroradiologists are reported in Figure 3.

Multivariate analysis of variance (MANOVA) was significant in discriminating among tumor subtypes (Pillai’s trace: *p* < 0.001). Post hoc Benjamini–Hochberg-corrected *t* tests showed that all DKI metrics were significantly different between medulloblastoma and pilocytic astrocytoma (*p* < 0.001 for all metrics). Moreover, RK, MK, and AK were significantly different between medulloblastoma and ependymoma (*p* < 0.012, *p* < 0.001, and *p* < 0.001 respectively), and AK, FA, and MD were significantly different between pilocytic astrocytoma and ependymoma (*p* < 0.048, *p* < 0.001, and *p* < 0.004, respectively). Boxplot differences across tumor subtypes for each metric are found in Figure 4. DKI-based findings to discriminate between pediatric posterior fossa tumors are reported in Figure 5A–L.

Post hoc Benjamini–Hochberg-corrected *t* tests showed all DKI metrics were significantly different between medulloblastoma and pilocytic astrocytoma (*p* < 0.001 for all metrics, asterisk). Moreover, RK, MK, and AK were significantly different between medulloblastoma and ependymoma (*p* < 0.012, *p* < 0.001, and *p* < 0.001, respectively, triangle), and AK, FA, and MD were significantly different between pilocytic astrocytoma and ependymoma (*p* < 0.048, *p* < 0.001, and *p* < 0.004, star, respectively). Thus, different specific DKI-derived diffusion metrics were useful in discriminating among specific tumor subtypes.

The RF machine learning analysis using the SMOTE dataset yielded an accuracy of 0.928, obtaining high AUC values for each tumor type (Table 2, Figure 6).

## 4. Discussion

In this work, we explored the repeatability (COV) of ROI and VOI DKI-derived diffusion measurements and analyzed whether VOI-based DKI was accurate in discriminating among pediatric posterior fossa tumors with machine learning techniques. Whole-tumor-based (VOI) measurements consistently demonstrated less variability (smaller COV) than both maximum area ROI and small ROI measurements, for all DKI-derived metrics. Moreover, VOI DKI metrics were accurate in discriminating among different tumor subtypes.

Prior to assessing DKI-derived metrics accuracy for PF tumor discrimination in our study, we tested whether these results would suffer from high inter-observer variability. This observation would imply a low repeatability and would reduce the confidence in the accuracy values we reported.

In line with this, we chose to segment the whole tumor volume and validate our choice by comparing COV between VOI-based and ROI-based measurements as well. Our assumption was that extending the analysis up to a volume of interest would reduce inter-observer variability, thus increasing the diagnostic accuracy of DKI metrics and allowing for better capture of lesion heterogeneity. We believe this assumption to be even more relevant in pediatric neuro-oncology studies, where high structural heterogeneity of each lesion and low patient numbers increase the need to maximize the reliable information, which can be obtained from each single case. Our results (volume < maximum area < small ROI variability across all metrics, Table 1, Figure 1) support our hypothesis. We obtained lower COV values for VOI-based measurements even as we tried to minimize the possible biases inherent with ROI-based measurements, by obtaining automatically the maximum ROI area instead of choosing it manually, and by selecting the highest values for the small ROI, instead of the mean values.

Diffusion stability is another aspect of the reliability of diffusion metrics, which we have not specifically explored in this paper. Interestingly, a recent multicenter study by Novak et al. investigated the classification of PF tumors by whole-volume tumor analysis based on diffusion-weighted imaging and machine learning [13]. Although the scope and methodology of the two papers differ widely, it is interesting to note that the boxplot distribution of ADC values from Figure 2 of their study and the boxplot distribution for MD (the mathematical equivalent of ADC) in Figure 3 of our study are very similar, with median values of ≈2.0 × 10^−3^ mm^2^ s^−1^ for PA, ≈1.5 for EP, and ≈ between 1 and 1.5 for MB. This may support the authors’ observations that diffusion measures tend to be reliable even when comparing heterogeneous acquisition protocols from different scanners, if whole-tumor volume analysis is used.

Discriminating between posterior fossa tumors has been a well-known and very researched topic.

Previous papers [3,5,6,7] reported that diffusion-weighted imaging was useful in discriminating among pediatric posterior fossa tumors. Others also explored DTI utility for pediatric tumors and found that DTI-derived ADC was highly correlated with tumor classification and cellularity [30]. However, some authors have reported limited utility in discriminating among specific posterior fossa tumor subtypes due to some overlap between ependymoma and MB [31].

In our cohort, the multivariate analysis was significant in discriminating among different tumor subtypes with DKI metrics (*p* < 0.001). Different specific DKI-derived metrics were useful in discriminating among specific tumor subtypes. We observed that ADC was helpful in differentiating between pilocytic astrocytomas and medulloblastomas and between pilocytic astrocytomas and ependymomas, as previously reported [3,5,6,7]. The results from our series suggest that ADC alone may not be sufficient for discriminating between medulloblastomas and ependymomas, although the aforementioned study by Novak et al. reasserted its usefulness [13]. We also found that FA presented significant differences between pilocytic astrocytoma and ependymoma and that all kurtosis metrics (MK, AK, RK) were significantly different and helped discriminate between medulloblastoma and ependymoma. These results could potentially highlight the usefulness of DKI, which allows us to obtain more information on water diffusion with one single sequence (Figure 4 and Figure 5), as reported previously in pediatric gliomas [21].

A recent paper by She et al. [32] retrospectively correlated DWI, IVIM-derived, and DKI metrics with histopathologic features and tested the accuracy of these metrics for grading in pediatric patients affected by brain tumors by means of both an ROI- and VOI-based approach. Although the authors found DKI to be a good estimator of tumor cellularity, DKI did not present a superior grading performance compared to conventional DWI metrics.

In our study, we obtained a high discriminative accuracy with the ML approach for DKI metrics, which suggests DKI may be useful for discriminating for PF tumors. This is also in line with our previous findings on DKI [21], which proved useful for grading pediatric gliomas and presented a high correlation with survival metrics.

In recent years, several studies based on ML classifiers have been tested for discriminating among pediatric PF tumor subtypes [10,11] and for developing a radiomic signature of PF ependymomas [12]. Most of these studies were based on conventional MRI sequences (mainly, T2-weighted and T1-weighted sequences), although some also incorporated ADC maps into their analysis. Here, we focused on the DKI sequence and did not investigate whether conventional sequences could help better discriminate among PF tumor subtypes. It would be interesting to assess how ML techniques perform on conventional sequences, conventional diffusion, and advanced diffusion techniques.

Our ML analysis pipeline included preprocessing with SMOTE and classification with RF. SMOTE analysis allows for maintaining the same distribution of the original data, avoiding the creation of different synthetic data which would likely bias the model [33]. The Random Forest classifier tends to perform better compared to other classifiers for multi-class classification purposes, as was the case in our cohort where classification between three types of lesions (MB, AP, EP) had to be performed. For this reason, we chose to use the combination of the SMOTE technique and RF, which is conceptually the same process employed by a Bagging classifier: First, it adds some sampling technique (over/undersampling) on the training data and then performs the classification tasks.

The Random Forest classifier is an ensemble learning method used for multi-class classification. It operates by constructing multiple decision trees during training and subsequently by outputting the class that is the mode of the classes predicted by individual trees. Each tree is trained on a random subset of the data and features, which helps in reducing overfitting and improving model robustness. This method is particularly effective for multi-class classification due to its ability to handle high-dimensional data and its robustness to noise.

Despite the promising results, this pilot study is based on a small number of patients. This limitation thus implies caution in the interpretation of the classification results, and validation on a wider cohort should be performed. Moreover, it is known that ependymomas are relatively less frequent compared to pilocytic astrocytomas and medulloblastomas [3]. For this reason, the class imbalance generated by the lower number of patients affected by ependymoma should also be considered when interpreting the results.

We segmented semiautomatically the solid tumor volume because we found the method relatively easy to implement in a clinical setting. Other segmentation methods, such as automated segmentation, may be more useful in further reducing segmentation time and inherent human bias and could help progress toward a completely automated analysis pipeline. Ideally, a completely automated imaging analysis tool would be the gold standard to extract data from MRI and analyze them directly, which would save time and be more suited to scale to multicenter projects. In this pilot study, however, we chose not to implement a fully automated image-based machine learning approach. We chose instead a very conservative approach where we first tested our whole-tumor volume imaging segmentation method, performed semiautomatically, before performing ML analyses.

The reason for our choice was twofold.

First, based on our experience, automated segmentation techniques may not always be as accurate as semi-automated or manual segmentation techniques performed by a trained medical specialist. One of the reasons why this aspect is not trivial is that adequate co-registration of different MRI sequences in medical imaging is a complex issue, and segmentation without expert supervision would have incurred the risk of including non-tumoral areas and inherently reducing the quality of the results.

Second, we believe this issue to be even more relevant when testing an advanced diffusion technique, such as DKI, that is just beginning to be validated in neurooncological pediatric patients [21].

Compared to conventional DWI, DKI has a significantly longer duration, therefore, implying pediatric patients are exposed to longer scan time. Further studies are needed to better assess the clinical benefits of DKI, although its multipurpose nature, which allows both quantitative analysis and tractography reconstructions for presurgical planning, partially mitigates this issue. Also, we believe that the future technical improvements in MRI, allowing more extensive use of multiband imaging and significant reduction in acquisition time of these diffusion sequences, will further increase their benefit.

Lastly, using other statistical measures other than just the mean of the tumor volume DKI measurements (e.g., histogram analysis [13] or radiomic-based ML approaches [10,11,12,13,34]) may better capture intra-tumoral heterogeneity and extract even more useful information from DKI in pediatric posterior fossa tumors.

## 5. Conclusions

In this pilot study, we analyzed the repeatability of volume-based (VOI) versus region-of-interest (ROI) diffusion kurtosis imaging (DKI)-derived diffusion measurements. Volume-based DKI measurements had consistently lower variability than region-of-interest-based measurements across all metrics, suggesting whole tumor volume measurements should be used when possible. Then, we explored whether volume-based DKI was accurate in discriminating among pediatric posterior fossa (PF) tumors, with machine learning (ML) techniques including random synthetic minority oversampling technique (SMOTE) and classification with Random Forest (RF). Volume-based DKI measurements were accurate in discriminating among posterior fossa tumors. Finally, a Random Forest machine learning classification algorithm was accurate in discriminating posterior fossa tumors on a SMOTE-generated dataset. In conclusion, in our study, a machine learning analysis of the whole tumor volume based on diffusion kurtosis imaging proved repeatable and useful for the discrimination of pediatric posterior fossa tumors.

## Figures and Tables

**Figure 1 cancers-16-02578-f001:**
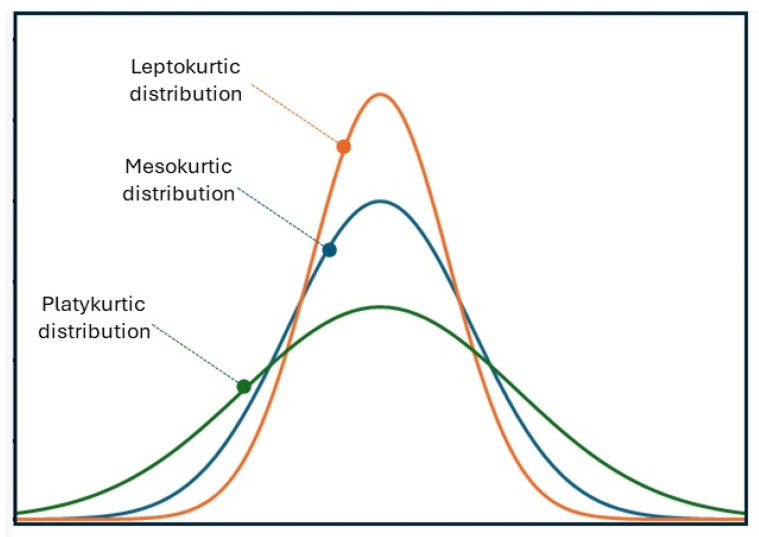
Simplified presentation of kurtosis in probability distributions. The platykurtic distribution (green curve, equivalent to higher kurtosis) presents a consistently larger tail distribution and a lower bell peak than the mesokurtic distribution (blue curve). By contrast, the leptokurtic distribution (orange curve, equivalent to lower kurtosis) presents a smaller tail and a higher peak than the mesokurtic distribution. This dimensionless metric was used to create the model for DKI and its derived metrics (MK, AK, RK), which measure non-Gaussianity in water diffusion in biological tissues.

**Figure 2 cancers-16-02578-f002:**
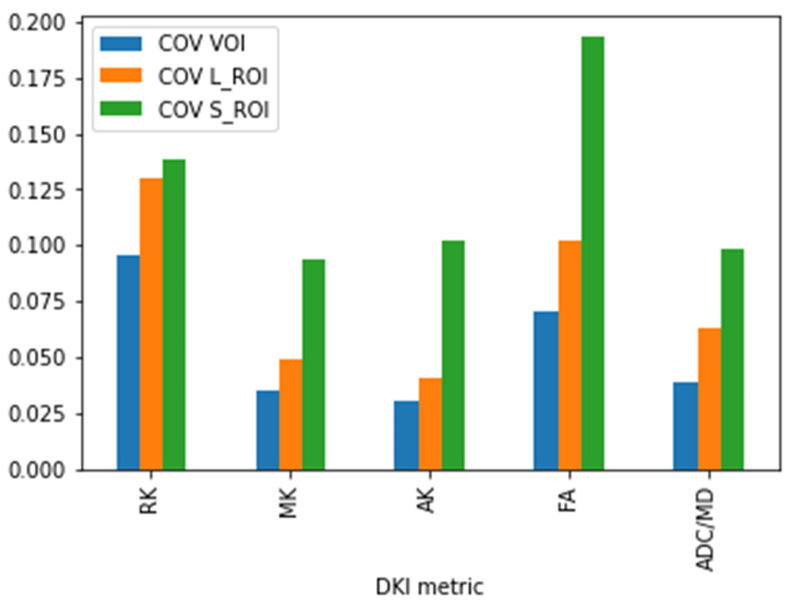
Bar graph coefficient of variation between the two readers. The volumetric segmentation (COV VOI, blue bars) presents consistently lower variability than the large region of interest (COV L_ROI, orange bars) and the small region of interest (COV S_ROI, green bars) across all metrics. The large ROI segmentation also presents a lower variability than the small ROI segmentation across all metrics.

**Figure 3 cancers-16-02578-f003:**
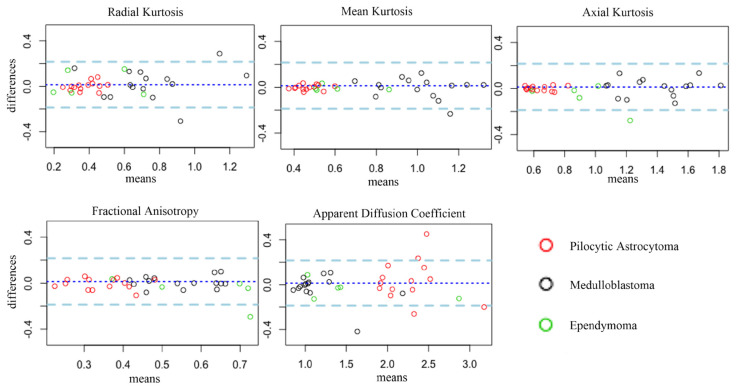
Bland–Altman plots of the mean versus differences of the tumoral volume measurements of the two readers for each DKI and DTI metric. Black dots represent patients affected by medulloblastoma (MB), red dots represent patients affected by pilocytic astrocytoma (PA), and green dots represent patients affected by ependymoma (EP). VOI-based measurements presented good to excellent repeatability across all metrics. All metrics were useful for the differentiation of PA (red dots) versus MB (black dots). RK, MK, and AK were useful for the differentiation of EP (green dots) versus MB (black dots). AK, FA, and ADC were useful for the differentiation of PA (red dots) versus EP (green dots).

**Figure 4 cancers-16-02578-f004:**
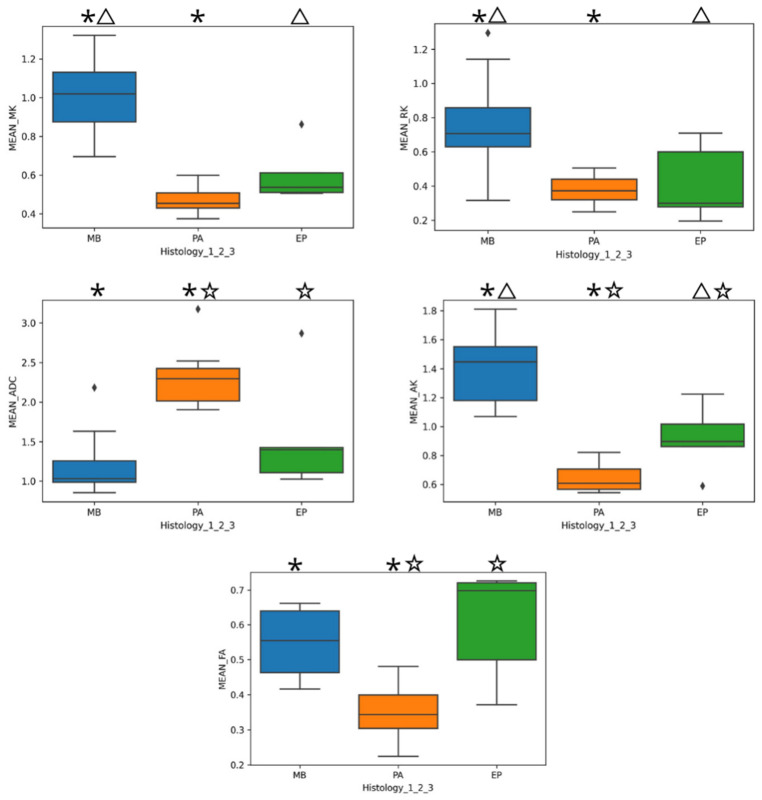
Boxplots of the mean of different DKI metrics (MK, RK, AK, ADC, and FA) related to each subgroup in the dataset. Significant differences among MB and PA, MB and EP, and EP and PA were marked with an asterisk, a triangle, and a star, respectively.

**Figure 5 cancers-16-02578-f005:**
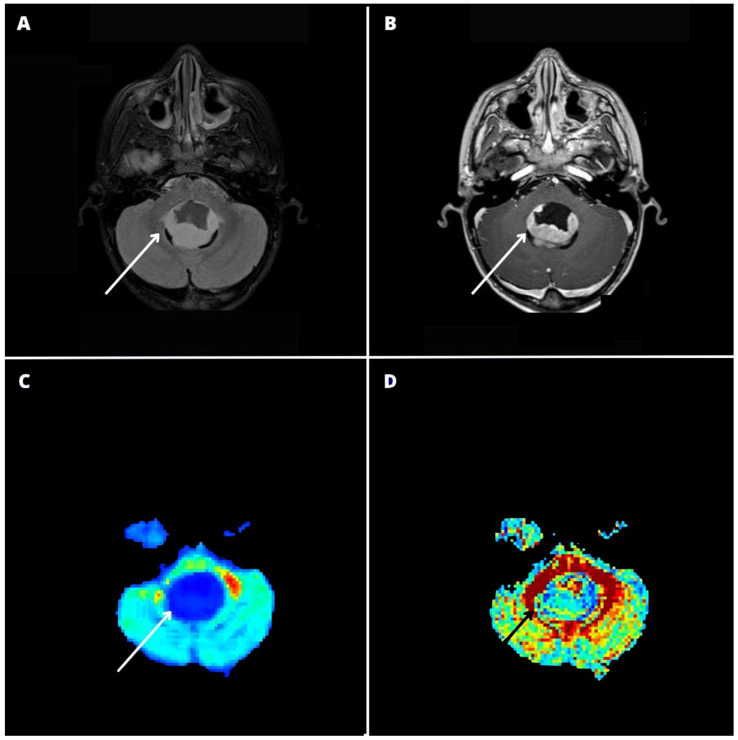
Diffusion kurtosis imaging-based findings to discriminate between pediatric posterior fossa tumors. MRI images are of a 7-year-old girl affected by a WHO grade I pilocytic astrocytoma (**A**–**D**), a 10-year-old boy affected by medulloblastoma (**E**–**H**), and an 8-year-old boy affected by a WHO grade II ependymoma (**I**–**L**). Native MPRAGE T1 post-contrast images (**B**,**F**,**J**) show three contrast-enhancing posterior fossa tumors located within the fourth ventricle, presenting a variable proportion of solid and cystic components. Qualitative evaluation of the mean kurtosis (MK) color maps (**C**,**G**,**K**) reveals significant differences between the lesions, with the pilocytic astrocytoma (PA) presenting relatively lower MK values ((**C**), white arrow), the ependymoma (EP) presenting intermediate MK values ((**K**), white arrow), and the medulloblastoma (MB) presenting high MK values compared with the other two lesions ((**G**), white arrow). Whole tumor labeling (green label) on the co-registered contrast-enhanced MPRAGE (**B**,**F**,**J**) and MK maps (**C**,**G**,**K**) yielded quantitative whole-tumor MK values of 0.40, 0.55, and 1.04, for PA, EP and MB respectively.

**Figure 6 cancers-16-02578-f006:**
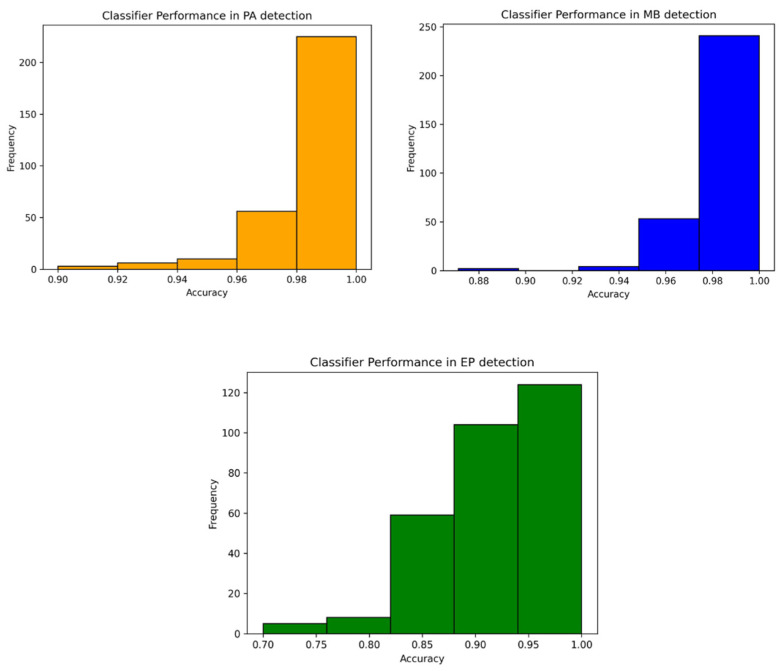
Histogram plots illustrating the performance of classifiers over 300 iterations in the machine learning analysis for the detection of the three tumor subgroups, pilocytic astrocytoma (PA, in orange), medulloblastoma (MB, in blue), and ependymoma (EP, in green). These boxplots show a more robust performance in MB and PA detection, compared to EP.

**Table 1 cancers-16-02578-t001:** Coefficient of variation between the two readers. The volumetric segmentation presents consistently lower variability than the large region of interest and the small region of interest across all metrics. The large ROI segmentation also presents a lower variability than the small ROI segmentation across all metrics.

DKI Metric	COV VOI	COV L_ROI	COV S_ROI
RK	9.70%	12.96%	13.84%
MK	3.53%	4.86%	9.36%
AK	3.02%	4.08%	10.23%
FA	6.74%	10.19%	19.3%
ADC/MD	3.91%	6.27%	9.8%

**Table 2 cancers-16-02578-t002:** Area under the curve (AUC) values and relative confidence intervals (CIs) for discriminating among tumoral subgroups (medulloblastoma = MB, pilocytic astrocytoma = PA, ependymoma = EP) in the synthetic minority oversampling technique (SMOTE) analysis.

Tumor	AUC	LOWER CI	UPPER CI
MB	0.96	0.96	1
PA	1	0.95	1
EP	0.98	0.84	1

## Data Availability

The data presented in this study are available upon request from the corresponding author. We provide the code for the analysis in the specific GitHub repository https://github.com/Francescodotta/Posterior_Fossa_Tumors_DKI/blob/main/Posterior_Fossa_DKI_Pipeline_Analysis.ipynb (accessed on 1 July 2024).

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
