# Peer review of "Machine Learning Analysis in Diffusion Kurtosis Imaging for Discriminating Pediatric Posterior Fossa Tumors: A Repeatability and Accuracy Pilot Study"

_cancers, 2024, doi:10.3390/cancers16142578_

Round 1

Reviewer 1 Report

Comments and Suggestions for Authors

In this article, the authors describe using machine learning-facilitated diffusion kurtosis imaging (DKI) to analyze MRI images. They compare the repeatability of their machine learning results and report that this method can accurately discriminate between different posterior fossa tumors. Below are some comments for improving the manuscript:

  1. Introduction of Techniques: The introduction should include a brief explanation of the fundamental differences between strategies such as region of interest (ROI) and volume of interest (VOI), as well as their relationship with DKI. Clarifying these aspects would greatly benefit the audience by providing a better understanding of the context and application of DKI.

  2. Clarification of Metrics: In line 97, the term “DTI metrics” should be explained when first introduced to avoid any confusion for the readers. A brief definition or explanation of these metrics will help in understanding their relevance and application in the study.

  3. Choice of Machine Learning Tools: Based on the data presented, machine learning classification was carried out solely on DKI metrics using a random forest classifier. It is unclear why the authors chose not to use image-based machine learning tools to analyze MRI images directly. Employing image-based techniques could potentially offer a more comprehensive analysis and enhance the robustness of their findings. Further clarification on this choice would strengthen the manuscript.

  4. Details on Machine Learning Algorithms: It is suggested that the authors provide more details about the machine learning algorithms used in the study. Ideally, they should offer access to the code to allow other researchers to examine the raw data and test the algorithm with their own experimental results. This transparency will facilitate reproducibility and validation of the findings within the scientific community.

Comments on the Quality of English Language

It is suggested that English should be edited by a native speaker.

Author Response

  1. Introduction of Techniques: The introduction should include a brief explanation of the fundamental differences between strategies such as region of interest (ROI) and volume of interest (VOI), as well as their relationship with DKI. Clarifying these aspects would greatly benefit the audience by providing a better understanding of the context and application of DKI.

    Thank you, we added some context in the introduction on differences between ROI and VOI measurements (lines 74-97). While DKI is a specific diffusion technique we are investigating in this paper, it may be argued that region of interest (ROI) vs volume of interest (VOI) difference is a more general topic and bears impact on how we analyze all imaging data, and not only those which are related to DKI. We added some context in the introduction and materials and methods section to make this clearer.

    We think this aspect is very relevant in imaging studies and especially pediatric neuro oncology studies (see also discussion, lines 343-359).

  2. Clarification of Metrics: In line 97, the term “DTI metrics” should be explained when first introduced to avoid any confusion for the readers. A brief definition or explanation of these metrics will help in understanding their relevance and application in the study.

    Thank you, in the introduction we added some context  to diffusion metrics, including DWI, DTI  and DKI, to better highlight their roles in clinical practice and in  the study. (lines 105-140)

  3. Choice of Machine Learning Tools: Based on the data presented, machine learning classification was carried out solely on DKI metrics using a random forest classifier. It is unclear why the authors chose not to use image-based machine learning tools to analyze MRI images directly. Employing image-based techniques could potentially offer a more comprehensive analysis and enhance the robustness of their findings. Further clarification on this choice would strengthen the manuscript.

    Thank you for this very important comment. Ideally, we would want a completely automated imaging analysis tool to extract data from MRI and analyze them directly, which would save time and be more scalable. In this paper however we chose not to implement a fully automated image-based machine learning approach. We chose instead a very conservative approach where we actually tested and proved our whole tumor volume imaging segmentation method, performed semiautomatically, to be more accurate and reliable than the alternative approaches that have been tried in literature.

    The reason for our approach was twofold.

    First, based on our experience, automated segmentation techniques are still not as accurate as semiautomated or manual segmentation techniques performed by a trained medical specialist. The reason why this aspect is not trivial is that coregistration of different MRI sequences in medical imaging is almost never perfect, and segmentation without expert supervision would have incurred the risk of including non-tumoral areas and inherently reduced the quality of the results.

    Second, we believe this issue to be even more relevant when testing an advanced diffusion technique, such as DKI, that is just beginning to be validated in neurooncological pediatric patients. As we believe this to be an important point, we added these concepts in the discussion.(lines 432-452).

  4. Details on Machine Learning Algorithms: It is suggested that the authors provide more details about the machine learning algorithms used in the study. Ideally, they should offer access to the code to allow other researchers to examine the raw data and test the algorithm with their own experimental results. This transparency will facilitate reproducibility and validation of the findings within the scientific community. 

    Thank you for this suggestion. We added some more context to the machine learning algorithm we implemented in the materials and methods and discussion section (lines 409-425).  Also, we  provide a public Github repository with the code used for the analysis in this pilot study (lines 261-262, https://github.com/Francescodotta/Posterior_Fossa_Tumors_DKI/blob/main/Posterior_Fossa_DKI_Pipeline_Analysis.ipynb). Due to privacy concerns, raw data are available upon request to the corresponding author.

Reviewer 2 Report

Comments and Suggestions for Authors

The authors presented an ML classification algorithm for discriminating posterior fossa (PF) tumors based on an imbalanced processing method, such as the Synthetic minority oversampling technique (SMOTE)—generated dataset. 

i.The presentation of the PF implementation is clear, and the manuscript is well-organized.

ii. It would be beneficial to include a detailed review of related work to help readers understand and summarize the pre-processing technique used to address class imbalance in the dataset.

iii. How does SMOTE compare to other imbalanced data pre-processing techniques such as Resampling (Oversampling and Undersampling), Balanced Bagging Classifier, and Threshold Moving? Why was SMOTE chosen?

iv. A detailed explanation of the ML classification algorithm would improve understanding.

v. Please provide a detailed explanation of the DKI metric to aid reader comprehension.

Comments on the Quality of English Language

NA

Author Response

i.The presentation of the PF implementation is clear, and the manuscript is well-organized.

Thank you. Below you can find our answers and revisions to your other comments.

  1. It would be beneficial to include a detailed review of related work to help readers understand and summarize the pre-processing technique used to address class imbalance in the dataset.

The pre-processing technique used to address the class imbalance in the dataset implied using the SMOTE technique to generate the same amount of data in each subgroup, in order to reduce the class imbalance problem. An alternative solution would be to increase the amount of data, and perform a random subsampling technique, which would allow the analysis of all real data. Such process would be impractical in our case study, since the class imbalance determined by the low number of patients with EP was better suited for an upsamling technique such as SMOTE. We added some context upon SMOTE and why we chose it over alternatives in the Discussion section.(lines 409-418)

iii. How does SMOTE compare to other imbalanced data pre-processing techniques such as Resampling (Oversampling and Undersampling), Balanced Bagging Classifier, and Threshold Moving? Why was SMOTE chosen?

Thank you for your question. As far as  data oversampling is concerned, SMOTE tries to maintain the same distribution of the data, so in such a way we maintain  the same distribution range of the real data without creating some synthetic data with different distributions, which would likely bias the model. The Random Forest Classifier tends to  perform better compared to other classifiers for multi-class classification purposes, so we chose to use the combination of SMOTE technique and Random Forest. This is also  the same process employed  by the Bagging classifier: since first it adds some sampling technique (over/undersampling) on the training data and then performs the classification tasks. We added this point in the discussion. (lines 409-418)

  1. A detailed explanation of the ML classification algorithm would improve understanding.
  2. Thank you for your question. After correcting for class imbalance with SMOTE, we implemented the classification algorithm Random forest. The Random Forest classifier is an ensemble learning method used for multi-class classification by constructing multiple decision trees during training. It operates by outputting the class that is the mode of the classes predicted by individual trees. Each tree is trained on a random subset of the data and features, which helps in reducing overfitting and improving model robustness. This method is particularly effective for multi-class classification due to its ability to handle high-dimensional data and its robustness to noise. After reviewing the paper, we added some sentences in the Materials and Methods (lines 232, 261-262) and in the Discussion (lines 409-425)section to clarify that.

  1. Please provide a detailed explanation of the DKI metric to aid reader comprehension.

Thank you, we added explanations of diffusion related metrics (DWI, DTI and DKI) in the introduction (lines 105-140) and a more detailed explanation of the DKI metric in a specific subsection (2.2) in the  materials and methods section.

Round 2

Reviewer 2 Report

Comments and Suggestions for Authors

The authors addressed all my comments. 

Comments on the Quality of English Language

NA